# Nested Hash Layer: A Plug-and-play Module for Multiple-length Hash Code Learning

## Abstract

Deep supervised hashing is essential for efficient storage and search in large-scale image retrieval. Traditional deep supervised hashing models generate single-length hash codes, but this creates a trade-off between efficiency and effectiveness for different code lengths. To find the optimal length for a task, multiple models must be trained, increasing time and computation. Furthermore, relationships between hash codes of different lengths are often ignored. To address these issues, we propose the Nested Hash Layer (NHL), a plug-and-play module for deep supervised hashing models. NHL generates hash codes of multiple lengths simultaneously in a nested structure. To resolve optimization conflicts from multiple learning objectives, we introduce a dominance-aware dynamic weighting strategy to adjust gradients. Additionally, we propose a long-short cascade self-distillation method, where long hash codes guide the learning of shorter ones, improving overall code quality. Experiments indicate that the NHL achieves an overall training speed improvement of approximately 5 to 8 times across various deep supervised hashing models and enhances the average performance of these models by about 3.4%.

## 1 Introduction

With the growing amount of visual data on the Internet, existing databases are becoming vast. To manage this data in large-scale image databases, hashing represents images as binary hash codes for efficient storage and search (Luo et al., 2023). Recently, deep supervised hashing has made significant progress by extracting deep features and using supervised signals to enhance hash code quality. As shown in the upper part of Figure 1a, the traditional approach involves using a deep neural network to extract features and a hash layer to generate hash codes[1]. The hash layer typically consists of a single-layer perceptron that maps features to the desired hash code length, followed by a binary operation (e.g., the signum function) to produce the final hash codes.

However, most deep supervised hashing models focus on generating hash codes of a specific length. This leads to two problems. First, Figure 1b shows the performance of four deep supervised hashing models (Liu et al., 2016; Wang et al., 2017; Cao et al., 2018; Wang et al., 2022) on the CIFAR-10 dataset at different code lengths. There is a clear trade-off between efficiency and effectiveness: shorter hash codes improve efficiency but reduce effectiveness, while longer hash codes enhance performance but increase storage and computational costs (Sun et al., 2023). Furthermore, this trade-off is not entirely inversely proportional, as seen with the DSH model, which suffers a notable performance drop at 128 bits, a phenomenon known as the dimension curse of hash code. As a result, multiple models must be trained for different code lengths to find the best fit for a specific task, greatly increasing training time and resource use (Wu et al., 2022). Second, since these models only produce single-length hash codes, they overlook the potential relationships between hash codes of different lengths. This raises a question: **is it possible to train a single hashing model capable of producing hash codes of multiple lengths?**

Code expansion and compression-oriented deep hashing methods address the variation in hash code lengths after training. They generate new codes either through code expansion (Mandal et al., 2019;

---

[1]Most deep hashing methods for image retrieval adhere to this paradigm, while some works (Shen et al., 2017; Jiang & Li, 2018; Chen et al., 2019; Wu et al., 2023) optimize the hash code in the database independently. Our work focuses on the former.

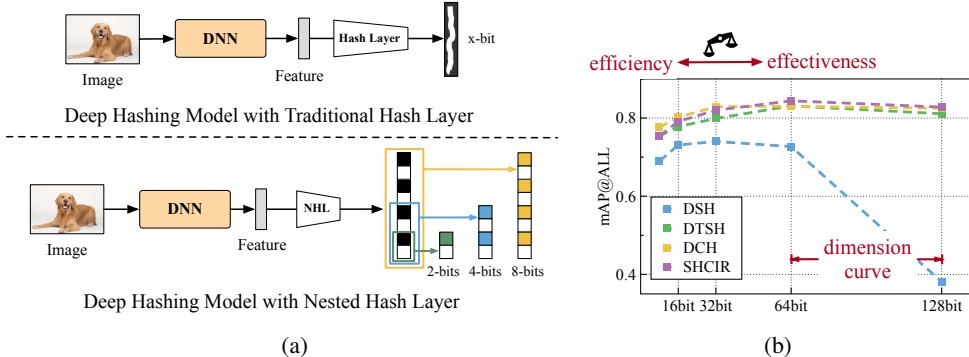

Figure 1: (a) A schematic comparison between traditional hash layer methods and our proposed NHL. The NHL can generate hash codes of multiple lengths simultaneously in a nested manner. (b) The performance of four deep supervised hashing models on the CIFAR-10 dataset highlights the uncertainty in effectiveness and efficiency under different code lengths.

Wu et al., 2022; 2024) or compression (Zhao et al., 2020). However, their primary focus is on learning a mapping after model training to convert existing hash codes into new ones. MAH (Luo et al., 2020) and SDMLH (Nie et al., 2022) involve the generation of hash codes with multiple lengths, but they rely on specifically designed models that cannot be generalized to other deep hashing models. Thus, designing a method capable of being widely applied to deep hashing models for generating hash codes of multiple lengths remains an unexplored area.

In this work, we propose the Nested Hash Layer (NHL), a plug-and-play module that replaces the traditional hash layer in deep supervised hashing models to generate hash codes of multiple lengths. **First**, we observe that deep supervised hashing models use the same backbone to extract features, regardless of code length. Additionally, longer hash codes can be seen as extensions of shorter ones. Based on these insights, the NHL is designed as shown in the lower part of Figure 1a to generate hash codes of different lengths in a nested manner, enabling multiple-length code generation in one model. **Second**, while the NHL combines objectives for multiple code lengths, conflicts may arise, as shorter hash codes are integral to longer ones. Thus, we introduce a Dominance-Aware Dynamic Weighting strategy. We define a "domination gradient" for each nested parameter, prioritizing the optimization of shorter hash codes. By monitoring parameter gradients, we dynamically adjust objective weights to align with the domination gradient and avoid conflicts. **Third**, unlike the traditional hash layer, NHL generates multiple-length hash codes. To further enhance code quality, we propose a Long-short Cascade Self-distillation method, leveraging the relationships in long hash codes to guide and improve shorter ones.

As evidenced by extensive experiments and analysis, NHL offers the following key advantages: (1) NHL achieves an overall training speed improvement of approximately 5 to 8 times across various deep supervised hashing models. (2) While ensuring faster training, NHL enhances the average performance of these models by about 3.4% and remarkably alleviates the dimensionality curse of hash codes. (3) NHL demonstrates exceptional flexibility in adapting to scenarios with multiple hash code length settings and different backbones.

## 2 RELATED WORK

### 2.1 DEEP SUPERVISED HASHING

Our work focuses on deep supervised hashing models, which can be roughly divided into pair-wise methods, ranking-based methods, and proxy-based methods. The objective of pair-wise methods (Liu et al., 2016; Zhu et al., 2016; 2017; Cao et al., 2018; Li et al., 2020; Zheng et al., 2020) is to ensure similar pairs have similar hash codes while dissimilar pairs have dissimilar hash codes. Ranking-based methods adopt ranking-based similarity-preserving loss terms. For instance, triplet loss (Wang et al., 2017; Liu et al., 2018) and list-wise loss (Cakir et al., 2019) are commonly used to maintain data ordering. Proxy-based methods (Yuan et al., 2020; Fan et al., 2020; Hoe et al., 2021;

Wang et al., 2022; 2023), also known as center-based methods, have emerged as a widely acclaimed approach recently. These methods first generate each category's proxies (or hash centers). Then, they force hash codes outputted from the network to approach corresponding proxies (or hash centers). Most of the current deep supervised hashing models only account for a single model with a specific code length. This limitation leads to slow training in practical applications due to the need to train multiple models with different hash code lengths. MAH (Luo et al., 2020) and SDMLH (Nie et al., 2022) attempt to solve this problem, but they rely on specifically designed models that cannot be generalized to other deep hashing models.

## 2.2 MULTI-TASK LEARNING

The NHL can be seen as a multi-task learning framework (Lee & Seok, 2023), where multiple related tasks are trained simultaneously using a shared model. Its primary function is to enable a single hash model to serve multiple learning objectives for different code lengths. A key aspect of multi-task learning is architecture design, including hard parameter sharing methods (Kokkinos, 2017; Bragman et al., 2019) and soft parameter sharing methods (Ruder et al., 2019; Gao et al., 2020; Liu et al., 2019). NHL only makes simple adjustments to the hash layer to accommodate various deep hashing models. MRL (Kusupati et al., 2022) partly inspired its basic structure, which generates representations of different lengths for multiple tasks. However, MRL does not address gradient conflicts or the relationships between representations of different lengths.

To address gradient or task conflicts, some methods re-weight the task losses based on specific criteria such as uncertainty (Kendall et al., 2018), gradient norm (Chen et al., 2018), or difficulty (Guo et al., 2018). Other methods leverage gradient information to modify the gradient on the parameter update procedure (Yu et al., 2020; Chen et al., 2020; Liu et al., 2021; Javaloy & Valera, 2022). Nevertheless, these multi-task learning methods assume the importance of different objectives is equivalent. In NHL, the weights of objectives are different because the short hash codes appear to hold greater significance. Resolving this problem remains a further exploration.

## 3 METHODOLOGY

### 3.1 PROBLEM DEFINITION

Given a database $X = \{x_i\}_{i=1}^{N}$ comprising $N$ images and $Y = \{y_i\}_{i=1}^{N}$ is the corresponding label set, deep supervised hashing targets to learn a hash function $f : x_i \mapsto h_i$ that maps each data $x_i \in X$ to a binary hash code $h_i \in \{-1, 1\}^b$, where $b$ denotes the length of hash code. This mapping aims to preserve the pairwise similarities between the images $x_i$ and $x_j$ in the Hamming space, characterized by the Hamming distance for hash codes $h_i$ and $h_j$. In this work, we aim to generate hash codes with $m$ code lengths $b = \{b_k\}_{k=1}^{m}$. Without loss of generality, we define $b_k < b_{k+1}$. Then, the image $x_i$ is mapped to $m$ different lengths of hash codes, denoted as $\{h_i^{(k)}\}_{k=1}^{m}$.

### 3.2 HASH CODE GENERATION

The process of acquiring hash codes of most deep supervised hashing methods is divided into two parts. First, a deep neural network is employed to extract the feature $v = \mathcal{F}(x) \in \mathbb{R}^l$ given the data $x_i \in X$, where $l$ is the dimension of $v$. Then, a hash layer is utilized to derive the hash code $h$. In most cases, the hash layer consists of a single-layer perceptron to map the data features to a length equivalent to that of the hash code, and an operation $\phi$ for acquiring the hash codes. The whole process to get the hash code $h$ can be formulated as follows:

$$h = f(x) = \phi(\mathcal{W}\mathcal{F}(x) + c), \tag{1}$$

where $\mathcal{W} \in R^{b \times l}$ and $c \in R^b$ are the parameters in the single-layer perception to be learned. Current deep hashing methods usually predefine a code length $b_k$ and then train a hash model $f_k$ accordingly. However, in practice, the selection of an appropriate code length depends on the specific task at hand, which means we need to train multiple deep hashing models $\{f_k\}_{k=1}^{m}$ for different code lengths and select the most suitable one. Such an approach will increase both the training time and computational resources required. To solve this problem, we introduce NHL to replace the original hash layer in deep hashing models. In the following section, we omit the bias $c$ and the operation $\phi$ for conciseness.

Figure 2: (a): The Nested Hash Layer (NHL) can generate $m$ (here, $m = 3$) hash codes with varying lengths in one training procedure. (b) The illustration of the Dominance-Aware Dynamic Weighting strategy. Taking the $\mathcal{W}^{(1)}$ as an example. (c) The Long-short Cascade Self-distillation transfer relationship from long hash codes to short hash codes.

## 3.3 BASIC STRUCTURE OF NESTED HASH LAYER

Although the predefined code lengths differ, the same backbone is employed for a specific deep hashing model. Inspired by this observation, we propose the basic structure of NHL to help deep hashing models generate hash codes with different code lengths in one training procedure.

As shown in Figure 2a, NHL uses a nested parameter $\{\mathcal{W}^{(k)}\}_{k=1}^{m}$ to achieve this goal without adding additional parameters to the neural network. The parameter $\mathcal{W}^{(k)} = \mathcal{W}_{[1:b_k]}^{(m)} \in \mathbb{R}^{l \times b_k}$ is in a nested structure, which means $\mathcal{W}^{(k)} \subset \mathcal{W}^{(k+1)}$. It uses the first $b_k$ vectors of the parameter $\mathcal{W}^{(m)} \in \mathbb{R}^{l \times b_m}$. We can obtain the hash codes with different lengths $\{h^{(k)}\}_{k=1}^{m}$ through $h^{(k)} = \phi(\mathcal{W}^{(k)} v^T)$. Then, we aim to minimize the following objective.

$$\mathcal{L} = \sum_{i=1}^{N} \sum_{k=1}^{m} \mathcal{L}_k(h_i^{(k)}, y_i; \theta_F, \mathcal{W}^{(k)}), \tag{2}$$

where $\theta_F$ is the parameter of backbone, and $\mathcal{L}_k$ is the objective of a specific deep hashing model for code length $b_k$. In most deep hashing models, $\mathcal{L}_k$ can be a combination of multiple objectives, such as the central similarity loss and quantization loss. As it simply involves adding the original objective of the deep hashing model, it does not alter the original optimization method. By minimizing Eq.(2), we force hash codes with different lengths to ensure their performance.

## 3.4 DOMINANCE-AWARE DYNAMIC WEIGHTING STRATEGY

Although basic NHL can generate hash codes with different lengths, we are unable to predict whether the gradients for different objectives $\{\mathcal{L}_k\}_{k=1}^{m}$ are mutually beneficial or detrimental. For example, in the left part of Figure 2b, the parameter $\mathcal{W}^{(1)}$ is updated by three gradients $g_1^{(1)} = \frac{\partial \mathcal{L}_1}{\partial \mathcal{W}^{(1)}}$, $g_2^{(1)} = \frac{\partial \mathcal{L}_2}{\partial \mathcal{W}^{(1)}}$, and $g_3^{(1)} = \frac{\partial \mathcal{L}_3}{\partial \mathcal{W}^{(1)}}$. Due to the impact of $g_2^{(1)}$, $g_3^{(1)}$, optimizing the parameters tends to proceed in a direction unfavourable to $g_1^{(1)}$ because the negative inner product between $g_1^{(1)}$ and $g_2^{(1)}$, $g_3^{(1)}$. However, The quality of the hash code $h^{(1)}$ is determined by objective $\mathcal{L}_1$, which updates $\mathcal{W}^{(1)}$ using the gradient $g_1^{(1)}$ based on the target's outcomes. Therefore, if the final optimization direction of $\mathcal{W}^{(1)}$ diverges from $g_1^{(1)}$, it is highly probable that such a deviation will lead to a deterioration in the quality of $h^{(1)}$ because the wrong optimization direction for it.

Some multi-task learning works (Yu et al., 2020; Chen et al., 2020; Liu et al., 2021; Javaloy & Valera, 2022; Guangyuan et al., 2022) propose modifying the gradient on the parameter update procedure to prevent gradient conflicts. However, there exists a difference between these multi-task learning settings and NHL. Multi-task learning treats diverse learning objectives as equally important, aiming to balance various learning objectives. In NHL, the objectives corresponding to shorter hash codes appear to hold greater significance, as shorter hash codes are shared by a larger number of longer hash codes. To address this problem, we propose a dominance-aware dynamic weighting strategy to adjust the weight $\alpha_k$ of each objective $\mathcal{L}_k$ by monitoring the gradients. Then the objective Eq. (2)

becomes follows:

$$\mathcal{L} = \sum_{i=1}^{N} \sum_{k=1}^{m} \alpha_k \mathcal{L}_k(h_i^{(k)}, y_i; \theta_F, \mathcal{W}^{(k)}). \tag{3}$$

Since shorter hash codes should be given higher optimization priority, we are motivated to introduce the following definitions.

**Definition 1** (Dominant gradient). Assume the gradient of $\mathcal{L}_i$ for $\mathcal{W}^{(k)}$ is denoted as $g_i^k = \frac{\partial \mathcal{L}_i}{\partial \mathcal{W}^{(k)}}$. We define $g_k^{(k)} = \frac{\partial \mathcal{L}_k}{\partial \mathcal{W}^{(k)}}$ is the dominant gradient, and $k = 1, 2..., m$. For example, $g_1^{(1)}$ is the dominant gradient in Figure 2b.

**Definition 2** (Anti-domination & Align-domination). Assume the gradient of $\mathcal{L}$ for $\mathcal{W}^{(k)}$ is $g^k = \frac{\partial \mathcal{L}}{\partial \mathcal{W}^{(k)}}$. We define anti-domination for the update of $\mathcal{W}^{(k)}$ if the inner product is negative between $g^{(k)}$ and the dominant gradient $g_k^{(k)}$, whereas a positive inner product is termed align-domination.

The dominant gradient $g_k^{(k)}$ serves as a guiding principle for the optimization of the parameter $\mathcal{W}^{(k)}$. Anti-domination and align-domination are thus employed to ascertain whether the update result of $\mathcal{W}^{(k)}$ is congruent with or divergent from the dominant gradient $g_k^{(k)}$. For example, the left part of Figure 2b shows that the update of $\mathcal{W}^{(1)}$ is anti-domination because the negative inner product between the gradient $g^{(1)}$ and $g_1^{(1)}$. We conducted an analysis to observe the occurrence of anti-domination as training progressed. Figure 3 depicts the likelihood of anti-domination occurring about parameter $\mathcal{W}^{(1)}$ at each epoch. These results reveal the probability of anti-domination steadily rises over time, eventually stabilizing at a level exceeding 90%. This trend signifies a growing prevalence of anti-domination scenarios as the training progresses.

Our goal is to avert anti-domination for each $\mathcal{W}^{(k)}$. We propose the following proposition:

**Proposition 1.** *Assume $\theta_{ij}^{(k)}$ is the angle between two gradients $g_i^{(k)}$ and $g_j^{(k)}$, and $\|\cdot\|$ denotes the Frobenius norm, if the following inequality holds:*

$$\alpha_k \|g_k^{(k)}\| + \sum_{k < i \leq m} \alpha_i cos\theta_{ik}^{(k)} \|g_i^{(k)}\| \geq 0, \tag{4}$$

*then the update of $\mathcal{W}^{(k)}$ is guaranteed to be align-domination.*

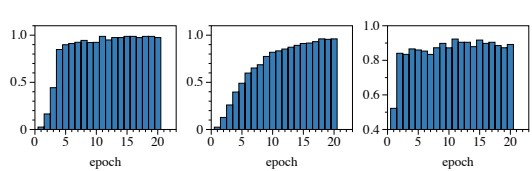

(a) CIFAR-10    (b) ImageNet100    (c) MSCOCO

Figure 3: The probability of anti-domination occurring on the parameter $\mathcal{W}^{(1)}$ at each epoch. We set the code lengths $m = 5$ and use CSQ as the deep hashing models on three datasets. This trend signifies a growing prevalence of anti-domination scenarios as the training progresses.

The proof is provided in Appendix B. However, the linear programming Eq.(4) is challenging to optimize and will incur additional time expenditure. Hence, we propose the following target and proposition:

**Proposition 2.** *If the following inequality holds for all $k < i \leq m$, then Eq. (4) also holds.*

$$\alpha_i cos\theta_{ik}^{(k)} \|g_i^{(k)}\| + \frac{\alpha_k}{m - k} \|g_k^{(k)}\| \geq 0. \tag{5}$$

We provide the proof in Appendix B and introduce a method to solve Eq. (5) in Appendix C. Here, for arbitrary $k \leq i \leq m$, we directly provide the results:

$$\alpha_i = min(\alpha_i^{(1)}, \alpha_i^{(2)}, ..., \alpha_i^{(i)}). \tag{6}$$

$$\alpha_i^{(k)} = \begin{cases} 1 & \text{if} \quad g_i^{(k)} \cdot g_k^{(k)} \geq 0 \\ \frac{\alpha_k}{k-m} \frac{\|g_k^{(k)}\|^2}{g_i^{(k)} g_k^{(k)}} & \text{if} \quad g_i^{(k)} \cdot g_k^{(k)} < 0. \end{cases} \tag{7}$$

In each training step, we dynamically compute the $\{\alpha_k\}_{k=1}^m$ using Eq. (6) and Eq. (7). Similar to (Chen et al., 2018), we don't consider the full network weights and focus on the parameter in NHL. The computation complex is $O(lb_m m^2)$, where $l$ is the dimension of data feature $v$ and $b_m$ is the longest code length. Our experiment shows that the additional training time for each step is around 11.15%.

### 3.5 LONG-SHORT CASCADE SELF-DISTILLATION

Unlike the traditional hash layer, the deep hashing model with NHL can generate hash codes of different lengths simultaneously. This leads us to explore the connections among these hash codes of different lengths. We observe a teacher-student relationship between long and short hash codes. Thus, as illustrated in Figure 2c, we propose the long-short cascade self-distillation method, which uses long hash codes to improve the performance of short hash codes in a cascading manner.

Specifically, for arbitrary image $x_i$, through the NHL we can get its corresponding hash codes $\{h_i^{(k)}\}_{k=1}^m$. Let $H_k = [h_1^{(k)}, h_2^{(k)}, ..., h_B^{(k)}] \in \{-1, 1\}^{B \times b_k}$ denote the matrix of hash codes with length $b_k$ in current training batch, and $B$ is the batch size. Then the self-distillation objectives can be formulated as:

$$\mathcal{L}_k^{lcs} = \frac{1}{B} \left\| \frac{h_i^{(k)} H_k^T}{\|h_i^{(k)} H_k^T\|} - \frac{h_i^{(k+1)} H_{(k+1)}^T}{\|h_i^{(k+1)} H_{(k+1)}^T\|} \right\|^2 . \tag{8}$$

Eq.(8) can be viewed as transferring the relationship between $h_i^{(k+1)}$ and other hash codes of length $b_{k+1}$ to the relationship between $h_i^{(k)}$ and other hash codes of length $b_k$. Besides, we stop the gradient propagation of the long hash codes $h_i^{(k+1)}$ and $H_{(k+1)}$ to ensure that the learning of relationships is unidirectional. In other words, we only allow the shorter hash codes to learn from the relationships of the longer hash codes. By introducing the long-short cascade self-distillation into the optimization procedure, the objective Eq.(3) becomes:

$$\mathcal{L} = \sum_{i=1}^{N} \sum_{k=1}^{m-1} \alpha_k (\mathcal{L}_k + \lambda \mathcal{L}_k^{lcs}) + \sum_{i=1}^{N} \alpha_m \mathcal{L}_m, \tag{9}$$

where $\lambda$ is a hyper-parameter. This method readily allows for expansion. For instance, one could explore the relationship between $h_k$ and $h_{k+a}$, where $a$ is an integer, but this is not the central concern of our work.

We renormalize the weights $\alpha_k$ in each step so that $\sum_{k=1}^m \alpha_k = m$ to decouple gradient re-weight from the global learning rate. Besides, in the training procedure, note that the minimum of $\mathcal{L}$ does not necessarily imply that each $\{\mathcal{L}_k\}_{k=1}^m$ is at its minimal value during the training process. Therefore, we propose a trick for our training procedure. Throughout the training, we monitor the value of each $\mathcal{L}_k$ and save the model parameters when each $\mathcal{L}_k$ reaches its minimum to output the corresponding hash codes $h^{(k)}$. We summarize the whole algorithm in Appendix D.

## 4 EXPERIMENTS

### 4.1 EXPERIMENT SETTINGS

We evaluated our method on three widely used datasets in deep hashing: CIFAR-10 (Krizhevsky et al., 2009), ImageNet100 (Deng et al., 2009), and MSCOCO (Lin et al., 2014). We compared our approach against several state-of-the-art deep supervised hashing baselines: DSH (Liu et al., 2016), DHN (Zhu et al., 2016), DTSH (Wang et al., 2017), LCDSH (Zhu et al., 2017), DCH (Cao et al., 2018), DBDH (Zheng et al., 2020), CSQ (Yuan et al., 2020), SHCIR (Wang et al., 2022), DPN (Fan et al., 2020), and MDSH (Wang et al., 2023). For all the above models, we adopted ResNet50 (He et al., 2016) as the backbone. The primary evaluation metric was mean Average Precision at top K ($mAP@K$). Unless otherwise specified, we set the hash code lengths $b \in \{8, 16, 32, 64, 128\}$ for the following experiments, as these lengths are prevalently used in previous works. Details regarding datasets, implementation, and evaluation settings are presented in Appendix A.

### 4.2 PERFORMANCE ON DEEP HASHING MODELS

In this experiment, we first compared the $mAP@K$ of different deep supervised hashing models on three datasets. Table 1 shows the results. We use "w/o NHL" to denote the deep hashing model without using NHL and use "w/ NHL" to denote the deep hashing model that uses NHL to replace the traditional hash layer. Besides, we use bold numbers to indicate statistically significant improvements when utilizing NHL compared to not using NHL, with $p < 0.05$ based on a two-tailed paired t-test.

Table 1: The mAP@K comparison results on CIFAR-10, ImageNet100, and MSCOCO datasets when different deep hashing models used the original hash layer (w/o NHL) or NHL (w/ NHL). We employ bold numbers to indicate statistically significant enhancements when utilizing NHL compared to when not using NHL, with $p < 0.05$ based on a two-tailed paired t-test.

| Data | Model | w/o NHL (The Original Model) | | | | | | w/ NHL | | | | | |
|---|---|---|---|---|---|---|---|---|---|---|---|---|---|
| | | 8 bits | 16 bits | 32 bits | 64 bits | 128 bits | avg. | 8 bits | 16 bits | 32 bits | 64 bits | 128 bits | avg. |
| CIFAR-10 (mAP@ALL) | DSH | 0.690 | 0.731 | 0.740 | 0.727 | 0.381 | 0.654 | **0.717** | 0.732 | 0.744 | **0.743** | **0.749** | **0.737** (+12.8%) |
| | DTSH | 0.754 | 0.778 | 0.799 | **0.831** | 0.811 | 0.745 | **0.766** | **0.790** | 0.802 | 0.822 | **0.836** | **0.771** (+3.51%) |
| | DHN | 0.718 | 0.765 | 0.813 | 0.837 | 0.853 | 0.797 | **0.739** | **0.776** | **0.824** | 0.835 | **0.866** | **0.808** (+1.38%) |
| | LCDSH | 0.715 | 0.771 | 0.817 | 0.826 | 0.854 | 0.797 | **0.775** | **0.799** | 0.825 | **0.839** | **0.868** | **0.821** (+3.08%) |
| | DCH | 0.776 | 0.802 | 0.829 | 0.830 | 0.825 | 0.812 | **0.787** | **0.810** | 0.833 | **0.843** | **0.844** | **0.823** (+1.36%) |
| | DBDH | 0.737 | 0.771 | 0.796 | 0.829 | 0.829 | 0.792 | **0.748** | **0.785** | 0.804 | 0.825 | **0.844** | **0.801** (+1.12%) |
| | CSQ | 0.762 | 0.786 | 0.798 | 0.798 | 0.807 | 0.790 | **0.792** | **0.802** | **0.818** | **0.828** | **0.838** | **0.816** (+3.21%) |
| | DPN | 0.703 | 0.757 | 0.790 | 0.804 | 0.819 | 0.775 | **0.729** | **0.765** | **0.795** | **0.826** | **0.824** | **0.788** (+1.71%) |
| | SHCIR | 0.754 | 0.791 | 0.820 | 0.844 | 0.828 | 0.807 | 0.758 | 0.797 | 0.824 | 0.849 | **0.848** | **0.815** (+0.99%) |
| | MDSH | 0.755 | 0.808 | 0.829 | 0.844 | 0.832 | 0.814 | **0.762** | 0.811 | **0.838** | **0.852** | **0.861** | **0.825** (+1.40%) |
| ImageNet100 (mAP@1000) | DSH | 0.703 | 0.808 | 0.827 | 0.828 | 0.822 | 0.797 | **0.755** | **0.816** | 0.829 | **0.838** | 0.841 | **0.817** (+2.41%) |
| | DTSH | 0.432 | 0.710 | 0.770 | 0.784 | **0.803** | 0.794 | **0.552** | 0.714 | 0.766 | **0.788** | 0.792 | **0.803** (+1.11%) |
| | LCDSH | 0.248 | 0.395 | 0.542 | 0.608 | 0.692 | 0.450 | **0.422** | **0.568** | **0.628** | **0.657** | 0.692 | **0.593** (+19.4%) |
| | DCH | 0.776 | 0.834 | 0.845 | 0.859 | 0.848 | 0.832 | **0.809** | **0.842** | **0.855** | 0.860 | **0.863** | **0.846** (+1.65%) |
| | CSQ | 0.456 | 0.822 | 0.860 | 0.877 | 0.878 | 0.778 | **0.495** | **0.825** | **0.873** | 0.880 | **0.882** | **0.787** (+1.59%) |
| | DPN | 0.436 | 0.827 | 0.864 | 0.870 | 0.877 | 0.775 | **0.487** | **0.829** | 0.860 | **0.877** | **0.881** | **0.787** (+1.50%) |
| | SHCIR | 0.789 | 0.861 | 0.879 | 0.883 | 0.881 | 0.858 | **0.798** | **0.881** | **0.889** | **0.893** | **0.898** | **0.872** (+1.63%) |
| | MDSH | 0.785 | 0.845 | 0.874 | **0.895** | 0.894 | 0.859 | **0.794** | **0.851** | 0.878 | 0.884 | 0.896 | 0.861 (+0.23%) |
| MSCOCO (mAP@5000) | DSH | 0.685 | 0.722 | 0.757 | 0.779 | 0.769 | 0.743 | **0.714** | **0.735** | **0.764** | 0.779 | **0.789** | **0.756** (+1.77%) |
| | DTSH | 0.706 | 0.770 | 0.810 | 0.823 | **0.831** | 0.788 | **0.751** | **0.793** | **0.819** | 0.826 | 0.823 | **0.803** (+1.86%) |
| | DHN | 0.659 | 0.751 | 0.786 | 0.810 | 0.832 | 0.768 | **0.724** | **0.760** | **0.794** | **0.819** | 0.837 | **0.787** (+2.47%) |
| | LCDSH | 0.687 | 0.769 | 0.787 | 0.825 | **0.836** | 0.781 | **0.713** | **0.773** | **0.794** | 0.820 | 0.828 | **0.786** (+0.55%) |
| | DCH | 0.695 | 0.756 | 0.762 | 0.777 | 0.734 | 0.745 | **0.723** | **0.769** | **0.786** | **0.788** | **0.789** | **0.771** (+3.51%) |
| | DBDH | 0.655 | 0.727 | 0.760 | 0.769 | 0.800 | 0.742 | **0.692** | **0.748** | **0.778** | **0.803** | **0.809** | **0.766** (+3.23%) |
| | CSQ | 0.596 | 0.750 | 0.847 | 0.877 | 0.871 | 0.788 | **0.659** | **0.778** | 0.847 | 0.878 | **0.881** | **0.809** (+2.58%) |
| | DPN | 0.575 | 0.757 | 0.828 | 0.862 | 0.863 | 0.777 | **0.638** | **0.769** | **0.837** | 0.863 | **0.872** | **0.796** (+2.44%) |

We can find the following observations: (i) Globally, the implementation of the NHL leads to an average improvement of 3.398% (3.619% in CIFAR-10, 4.364% in ImageNet100, and 2.158% in MSCOCO). Besides, there are 72% of cases that achieve a significant performance boost based on the two-tailed paired t-test. Conversely, only a few cases achieve a decline, with most drops of 1.37% occurring in the DTSH model when NHL is applied to the ImageNet100 dataset using a 128-bit code. Thus, we can demonstrate that NHL can yield significant improvements in the majority of cases. (ii) Deep hashing models with NHL improve significantly when the hash code length is short in some datasets. For example, in the case of 8-bit, employing NHL can increase 18.7% and 7.32% enhancement on ImageNet100 and MSCOCO datasets, respectively. (iii) It is delightful to note that NHL can address the dimensionality curse of hash code, signifying that as the code length expands to a certain dimension, the code quality commences to deteriorate in some deep hashing models. For example, without NHL, the quality of hash codes in DSH experiences a marked decline when transitioning from 64 bits to 128 bits on the CIFAR-10 dataset. In contrast, with the incorporation of the NHL, this result undergoes a substantial improvement.

Besides, to analyze the influence of each component in the NHL, we conducted an ablation study on these models to investigate their impact. We devised several variants for the NHL, namely (i) NHL-basic: directly use E.q (2) to optimize the deep hashing model, (ii) NHL w/o D: without using

Table 2: The comparison of average mAP@K results with the original model is shown for NHL variants.

| Data | NHL-basic | NHL w/o D | NHL w/o L | w/ NHL |
|---|---|---|---|---|
| CIFAR-10 | +1.088% | +1.316% | +2.097% | +3.619% |
| ImageNet100 | +2.209% | +2.823% | +2.434% | +4.364% |
| MSCOCO | +0.421% | +0.859% | +0.978% | +2.158% |
| avg. | +1.228% | +1.639% | +1.856% | +3.398% |

the dominance-aware dynamic weighting strategy, (iii) NHL w/o L: without using the long-short cascade self-distillation. Table 2 presents the average performance changes across datasets for the models mentioned above. Overall, implementing NHL-basic, NHL w/o D, and NHL w/o L results in average performance improvements of 1.228%, 1.639%, and 1.856%, respectively. Employing only the dominance-aware dynamic weighting strategy (NHL w/o L) achieves the most significant improvement, highlighting the critical role of gradient optimization in this context. In Appendix F, we present the details of various deep hashing models when utilizing the different variants of NHL.

## 4.3 EFFICIENCY ANALYSIS

In this experiment, we evaluated the deep hashing model's training time and memory usage. We recorded the total training time for the hashing model of five code lengths and recorded the maximal

Table 3: We evaluate efficiency on three datasets by recording the total training time for the deep hashing model with five code lengths and the maximum memory usage.

| Data | Model | Time (hours) | | Memory (GiB) | |
|---|---|---|---|---|---|
| | | w/o NHL | w/ NHL | w/o NHL | w/ NHL |
| CIFAR-10 | CSQ | 0.455 | 0.083 (5.48×) | 12.822 | 12.868 |
| | DCH | 0.543 | 0.085 (6.39×) | 12.820 | 12.836 |
| ImageNet100 | CSQ | 4.520 | 0.664 (6.81×) | 12.923 | 13.139 |
| | DCH | 6.025 | 1.161 (5.19×) | 12.914 | 12.956 |
| MSCOCO | CSQ | 1.019 | 0.202 (5.05×) | 12.906 | 13.067 |
| | DCH | 5.475 | 0.981 (5.58×) | 12.886 | 12.926 |

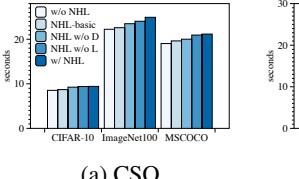

(a) CSQ  (b) DCH

Figure 4: The average training time per epoch for CSQ and DCH across three datasets, using the traditional hash layer, NHL, or NHL variants.

memory usage. Table 3 displays the results, where we selected CSQ and DCH as the deep hashing models. These results demonstrate that the employment of the NHL incurs negligible additional memory expenses. This is attributed to the fact that during the training process, the primary memory usage stems from the parameters of the neural network, while the additional memory occupied by the target loss is relatively minimal. Meanwhile, the incorporation of the NHL can significantly enhance the overall training speed. It achieved an average training speedup of $5.94\times$, $6\times$, and $5.31\times$ on the CIFAR-10, ImageNet-100, and MSCOCO datasets, respectively, across the two deep hashing models. In conjunction with the conclusions drawn from the previous experiment, this evidences that NHL can effectively expedite the training procedure without compromising the quality of hash codes.

Besides, we further analyze the average time per epoch during training across different NHL variants. Figure 4a and Figure 4b display the results of CSQ and DCH. Compared to deep hashing models without NHL (w/o NHL), incorporating NHL-basic, NHL w/o D, NHL w/o L, and w/ NHL led to a modest increase in training time of just 3.37%, 6.87%, 11.15%, and 13.75%, respectively.

### 4.4 Module Analysis

In this experiment, we conducted a comprehensive analysis of the Nested Hash Layer (NHL) from multiple perspectives, including (i) hyperparameter analysis and (ii) various code length settings. We utilize CSQ as the deep hashing model for the subsequent analysis. Additionally, we validated the performance of the Nested Hash Layer (NHL) under different backbone extractors, including other variants of ResNet and different architectures of Vision Transformers (ViT) (Dosovitskiy et al., 2020). For further details, please refer to Appendix E.2.

#### 4.4.1 Parameter Sensitivity

In Eq.(9), $\lambda$ serves as a hyperparameter that balances two objectives. We evaluated its values from $\{10^1, 10^0, 10^{-1}, 10^{-2}, 10^{-3}\}$ to calculate the $mAP@K$ across three datasets. Figure 5 illustrates the results, highlighting that the performance of NHL demonstrates overall robustness to changes in $\lambda$. Generally, the optimal value is achieved when $\lambda = 1$. Additionally, we conducted a parameter analysis on the learning rate, which is presented in Appendix E.1. The analysis indicates that the optimal range for the learning rate is $\{10^{-4}, 10^{-5}\}$.

#### 4.4.2 More Code Length Settings

This section explores the results under a broader range of code length settings. We established three scenarios for code length: **Case 1** sets a code length at every 32-bit interval, that is, $b = \{32 \times k\}_{k=1}^4$ and $m = 4$. **Case 2** sets a code length at every 16-bit interval, that is, $b = \{16 \times k\}_{k=1}^8$ and $m = 8$. **Case 3** sets a code length at every 8-bit interval, that is, $b = \{8 \times k\}_{k=1}^{16}$ and $m = 16$. Figure 6 presents the corresponding results. Here, the blue bars represent the average ratio of $mAP@K$ at various lengths with and without using NHL. The yellow bars indicate the time cost ratio to complete training with and without using NHL. We observe that even with different code length settings, the use of NHL ensures a reduction in overall training time and improves the code quality. Moreover, it is noteworthy that the total training time does not monotonically increase with the number of output code lengths. For instance, the efficiency enhancement ratio in Case 3 is not as high as in Case 2. This is attributed to the requirement for the model to undergo more training iterations in Case 3 to ensure favorable outcomes across a greater number of code lengths.

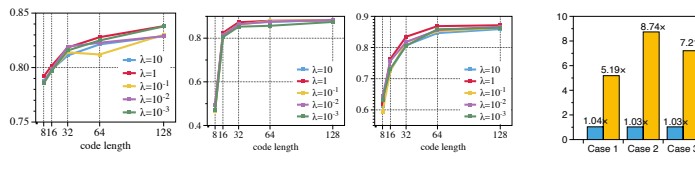

(a) CIFAR-10    (b) ImageNet100    (c) MSCOCO          (a) CIFAR-10    (b) ImageNet100    (c) MSCOCO

Figure 5: The $mAP@K$ results across three datasets under varying $\lambda$ values demonstrate that NHL is robust to $\lambda$ to some extent.

Figure 6: Impact of NHL on average mAP@K ratio (blue bars) and training time cost ratio (yellow bars) across various code length settings.

Table 4: The $mAP@K$ results across three datasets using different gradient conflict resolution strategies, with CSQ as the deep semantic hashing model. The best results are highlighted in bold, while the second-best results are underlined.

| Method | CIFAR-10 | | | | | ImageNet100 | | | | | MSCOCO | | | | |
|---|---|---|---|---|---|---|---|---|---|---|---|---|---|---|---|
| | 8 bits | 16 bits | 32 bits | 64 bits | 128 bits | 8 bits | 16 bits | 32 bits | 64 bits | 128 bits | 8 bits | 16 bits | 32 bits | 64 bits | 128 bits |
| NHL w/o D | 0.707 | 0.773 | 0.815 | 0.823 | 0.831 | 0.451 | 0.803 | 0.846 | 0.873 | **0.884** | 0.638 | 0.769 | 0.822 | 0.866 | 0.877 |
| + PCGrad | 0.742 | 0.781 | 0.802 | 0.815 | 0.830 | 0.467 | 0.814 | 0.852 | 0.870 | 0.862 | 0.632 | 0.772 | 0.823 | 0.852 | 0.872 |
| + GradDrop | 0.739 | 0.764 | 0.805 | 0.802 | 0.828 | 0.461 | 0.815 | 0.859 | 0.872 | 0.861 | 0.636 | 0.776 | 0.827 | 0.861 | 0.875 |
| + CAGrad | 0.720 | 0.775 | 0.793 | 0.811 | 0.815 | 0.455 | 0.815 | 0.867 | 0.865 | 0.873 | 0.639 | 0.770 | 0.825 | 0.858 | 0.869 |
| + RotoGrad | 0.735 | 0.762 | 0.813 | 0.819 | 0.822 | 0.473 | 0.808 | 0.863 | 0.874 | 0.877 | 0.641 | **0.778** | 0.830 | 0.859 | 0.871 |
| NHL | **0.792** | **0.802** | **0.818** | **0.828** | **0.838** | 0.495 | **0.825** | **0.873** | **0.880** | 0.882 | **0.659** | 0.778 | **0.847** | **0.878** | **0.881** |
| MAH | 0.636 | 0.649 | 0.683 | 0.705 | 0.714 | **0.628** | 0.652 | 0.689 | 0.692 | 0.691 | 0.567 | 0.573 | 0.589 | 0.599 | 0.612 |
| SDMLH | 0.617 | 0.684 | 0.723 | 0.748 | 0.763 | 0.526 | 0.557 | 0.618 | 0.623 | 0.644 | 0.602 | 0.646 | 0.737 | 0.761 | 0.814 |

### 4.5 COMPARED WITH GRADIENT CONFLICTS METHODS AND MULTI-LENGTH HASHING

In this section, we compare our proposed dominance-aware dynamic weighting strategy with other methods aimed at resolving gradient conflicts and two multi-length hashing. Specifically, we evaluate classic gradient conflicts approaches including PCGrad (Yu et al., 2020), GradDrop (Chen et al., 2020), CAGrad (Liu et al., 2021), and RotoGrad (Javaloy & Valera, 2022). The comparison is conducted by replacing the dominance-aware dynamic weighting strategy in our NHL method with the gradient update strategies of these methods and then calculating the $mAP@K$ results across three datasets, using CSQ as the deep hashing model. Besides, we also compared two multi-length hashing models MAH (Luo et al., 2020) and SDMLH (Nie et al., 2022). Table 4 presents the results. It can be observed that, compared to other gradient conflict resolution approaches, our method achieves the best results in more cases. Furthermore, we notice that incorporating these methods often results in negative effects compared to not using any gradient conflict resolution strategy (NHL w/o D). We believe this is due to the nested structure of our parameters, which these classical methods fail to account for adequately. Additionally, the performance of the two multi-code length hashing models, MAH and SDMLH, under most scenarios lags behind that of the CSQ model combined with NHL. This underscores the importance of designing a plug-and-play module, as it allows seamless integration with more advanced deep hashing models to achieve superior retrieval results.

## 5 CONCLUSION

In this paper, we introduce the plug-and-play module NHL for deep hashing models. NHL allows these models to generate hash codes of different lengths simultaneously, simplifying training and reducing computational load. Additionally, the dominance-aware dynamic weighting strategy and long-short cascade self-distillation enhance NHL's effectiveness. We performed extensive experiments on three datasets to assess NHL's performance. The results show that NHL speeds up training while maintaining or improving retrieval effectiveness across various deep supervised hashing models. Our work mainly focuses on common symmetric deep supervised hashing methods, where both the database and query data use the same deep hashing network for generating hash codes. In contrast, NHL is limited to asymmetric deep supervised hashing methods (Shen et al., 2017; Jiang & Li, 2018; Chen et al., 2019; Wu et al., 2023). They use deep neural networks only for processing query, while direct or indirect optimization of hash codes in the database. Thus, simply changing the hash layer isn't enough for these methods. We believe that adding optimization designs for the database hash codes could be a promising solution.

ETHICS STATEMENT

We confirm that our work adheres to the ICLR Code of Ethics. All datasets used in this research are publicly available and are appropriately cited in the paper. Our study does not involve human subjects, the collection of private user data, or the generation of personally identifiable or sensitive content. Therefore, our work raises no direct ethical concerns regarding privacy, security, or fairness.

REPRODUCIBILITY STATEMENT

We are committed to ensuring the reproducibility of our work. The source code is included in the supplementary materials and will be made publicly available upon publication. All datasets used in this paper are publicly available.

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

## A    EXPERIMENTAL SETTINGS

### A.1    DATASETS

We conducted experiments on three widely used datasets in deep hashing for evaluation. **CIFAR-10** (Krizhevsky et al., 2009) consists of 60,000 images from 10 classes. Following (Cao et al., 2018), we randomly select 1,000 images per class as the query set, and 500 images per class as the training set, and use all remaining images as the database. **ImageNet100** is a subset of ImageNet (Deng et al., 2009) with 100 classes. We follow the settings from (Fan et al., 2020) and randomly select 100 categories. Then, we use all the images of these categories in the training set as the database and the images in the validation set as the queries. Furthermore, we randomly select 13,000 as the training images from the database. **MSCOCO** (Lin et al., 2014) is a large-scale multi-label dataset. We consider a subset of 122,218 images from 80 categories, as in previous works (Qiu et al., 2021). We randomly select 5,000 images from the subset as the query set and use the remaining images as the database. For training, we randomly select 10,000 images from the database. As in most deep hashing settings, two samples are viewed as similar if they correspond to the same label on CIFAR-10 and ImageNet100. For multi-label datasets MSCOCO, two samples are considered similar if they share at least one common label.

### A.2    BASELINES AND EVALUATION METRIC

We considered the following deep supervised hashing models: DSH (Liu et al., 2016), DHN (Zhu et al., 2016), DTSH (Wang et al., 2017), LCDSH (Zhu et al., 2017), DCH (Cao et al., 2018), DBDH (Zheng et al., 2020), CSQ (Yuan et al., 2020), SHCIR (Wang et al., 2022), DPN (Fan et al., 2020), and MDSH (Wang et al., 2023). For all the above models, we uniformly adopted ResNet50 (He et al., 2016) as the backbone to extract 2048-dimensional image features.

We employed the mean Average Precision at the top K ($mAP@K$) as the evaluation metric. Specifically, we utilized $mAP@ALL$ for CIFAR-10, $mAP@5000$ for MSCOCO, and $mAP@1000$ for ImageNet100, following the settings used in previous studies (Qiu et al., 2021; Fan et al., 2020). Unless otherwise specified, we set the hash code lengths $b \in \{8, 16, 32, 64, 128\}$ for the following experiments, as these lengths are prevalently used in previous works.

### A.3    TRAINING DETAILS

For the deep hashing models we adopted, we endeavored to implement all models using PyTorch, based on the code repositories provided in the original papers and the implementation details described therein. The experiments were conducted on a Linux server equipped with 8 NVIDIA GeForce RTX 4090 GPUs. For each model (including those integrated with NHL), a single NVIDIA GeForce RTX 4090 GPU was utilized for both training and testing. Among these models, DHN and DBDH failed to produce valid results on ImageNet100 due to the absence of hyperparameter settings in the original papers, and our grid search method was unable to identify suitable parameters. Additionally, the SHCIR and MDSH models did not propose methods for handling multi-label datasets (e.g., MSCOCO) in their original publications. The batch size $B$ was set to 64. When applying NHL to the deep hashing models, we employed the Adam optimizer (Kingma & Ba, 2014) and selected the learning rate from $\{10^{-4}, 10^{-5}\}$. The hyperparameter $\lambda$ was set to 1. A grid search method was conducted across different scenarios to identify the optimal combination.

## B    PROOFS

**Proposition 1.** *Assume $\theta_{ij}^{(k)}$ is the angle between two gradients $g_i^{(k)}$ and $g_j^{(k)}$, and $\|\cdot\|$ denotes the Frobenius norm, if the following inequality holds:*

$$\alpha_k \|g_k^{(k)}\| + \sum_{k < i \leq m} \alpha_i cos\theta_{ik}^{(k)} \|g_i^{(k)}\| \geq 0, \tag{10}$$

*then the update of $\mathcal{W}^{(k)}$ is guaranteed to be align-domination.*

*Proof.* Align-domination is satisfied if the inner product between the total gradient $g^{(k)} = \sum_{i=k}^{m} \alpha_i g_i^{(k)}$ and the dominant gradient $g_k^{(k)}$ is non-negative:

$$\langle g^{(k)}, g_k^{(k)} \rangle \geq 0.$$

Expanding the inner product:

$$\langle g^{(k)}, g_k^{(k)} \rangle = \alpha_k \|g_k^{(k)}\|^2 + \sum_{k < i \leq m} \alpha_i \|g_i^{(k)}\| \|g_k^{(k)}\| \cos \theta_{ik}^{(k)}.$$

Dividing through by $\|g_k^{(k)}\|$ (assuming $\|g_k^{(k)}\| > 0$), we obtain:

$$\alpha_k \|g_k^{(k)}\| + \sum_{k < i \leq m} \alpha_i \cos \theta_{ik}^{(k)} \|g_i^{(k)}\| \geq 0.$$

Thus, if the inequality holds, the inner product $\langle g^{(k)}, g_k^{(k)} \rangle \geq 0$, ensuring that the total gradient $g^{(k)}$ is aligned with the dominant gradient $g_k^{(k)}$. This confirms align-domination for the update of $\mathcal{W}^{(k)}$. $\square$

**Proposition 2.** *If the following inequality holds for all $k < i \leq m$, then Eq. (10) also holds.*

$$\alpha_i cos\theta_{ik}^{(k)} \|g_i^{(k)}\| + \frac{\alpha_k}{m-k} \|g_k^{(k)}\| \geq 0. \tag{11}$$

*Proof.* For any fixed $k$ and $k < i \leq m$, Eq. (11) can be rewritten as:

$$\alpha_i \cos \theta_{ik}^{(k)} \|g_i^{(k)}\| \geq -\frac{\alpha_k}{m-k} \|g_k^{(k)}\|. \tag{12}$$

This inequality establishes a lower bound on the contribution of each term $\alpha_i \cos \theta_{ik}^{(k)} \|g_i^{(k)}\|$ for $k < i \leq m$. Then, summing Eq. (12) over all $k < i \leq m$, we obtain:

$$\sum_{k < i \leq m} \alpha_i \cos \theta_{ik}^{(k)} \|g_i^{(k)}\| \geq \sum_{k < i \leq m} -\frac{\alpha_k}{m-k} \|g_k^{(k)}\|. \tag{13}$$

Since there are exactly $m - k$ terms in the summation over $k < i \leq m$, the right-hand side simplifies to:

$$\sum_{k < i \leq m} -\frac{\alpha_k}{m-k} \|g_k^{(k)}\| = -(m-k) \cdot \frac{\alpha_k}{m-k} \|g_k^{(k)}\| = -\alpha_k \|g_k^{(k)}\|. \tag{14}$$

Thus, we have:

$$\alpha_k \|g_k^{(k)}\| + \sum_{k < i \leq m} \alpha_i \cos \theta_{ik}^{(k)} \|g_i^{(k)}\| \geq 0. \tag{15}$$

This proves that Eq. (10) holds whenever Eq. (11) holds for all $k < i \leq m$. $\square$

## C  THE METHOD TO SOLVE LINEAR PROGRAMMING PROBLEM EQ. 5

In the dominance-aware dynamic weighting strategy, we propose the following target for the objective weights $\{\alpha_k\}_{k=1}^m$:

$$\alpha_i cos\theta_{ik}^{(k)} \|g_i^{(k)}\| + \frac{\alpha_k}{m-k} \|g_k^{(k)}\| \geq 0; \ k \leq i \leq m. \tag{16}$$

This section describes how to solve it. Without loss of generality, we first set $\alpha_1 = 1$ for the shortest code' objective $\mathcal{L}_1$, as normalization can subsequently be applied. Then we introduce $\alpha_i^{(k)}$ denote only consider to ensure that $L_i$ and $L_k$ satisfy Eq. (16) on $\mathcal{W}^{(k)}$. Using $cos\theta_{ik}^{(k)} = \frac{g_i^{(k)} g_k^{(k)}}{\|g_i^{(k)}\| \|g_k^{(k)}\|}$ and re-arranging terms, we then get:

$$\alpha_i^{(k)}(-g_i^{(k)} g_k^{(k)}) \leq \frac{\alpha_k}{m-k} \|g_k^{(k)}\|^2; \ k \leq i \leq m. \tag{17}$$

---

**Algorithm 1** The training algorithm with NHL

---

**Input:** training samples $X = \{x_1, x_2, ...x_N\}$, the hyper-parameters $\lambda$.
 1: Initialization: the parameter of the deep hashing model $\{\theta_F, \mathcal{W}\}$, $\alpha_k = 1$ for $\forall k$.
 2: **repeat**
 3:   draw a mini-batch $\{x_1, x_2, ..., x_B\}$ from $X$ to compute $\{\mathcal{L}_k\}_{k=1}^m$ using standard forward propagation algorithm
 4:   **for** each $k \in \{1, 2, ...., m\}$ **do**
 5:     **for** each $i \in \{1, 2, ...., m\}$ **do**
 6:       obtain $g_i^{(k)}$ by computing standard gradients $g_i^{(k)} = \frac{\partial \mathcal{L}_i}{\partial \mathcal{W}^{(k)}}$ {Only calculating the gradients on $\{\mathcal{W}^{(k)}\}_{k=1}^m$}
 7:     **end for**
 8:   **end for**
 9:   compute $\{\alpha_k\}_{k=1}^m$ by Eq. (7) and Eq. (6)
 10:   renormalize $\{\alpha_k\}_{k=1}^m$ so that $\sum_{k=1}^m \alpha_k = m$
 11:   update parameters of the deep hashing model by minimizing Eq. (9) using the standard backpropagation algorithm
 12:   **if** achieved a smaller $\mathcal{L}_k$ **then**
 13:     record the current model parameters $\theta_F^{(k)}, \mathcal{W}^{(k)}$ for the model of length $b_k$.
 14:   **end if**
 15: **until** converged
**Output:** parameters of deep hashing model $\{\theta_F^{(k)}\}_{k=1}^m$ and $\{\mathcal{W}^{(k)}\}_{k=1}^m$

---

If $g_i^{(k)} g_k^{(k)} \geq 0$, because $\alpha_j > 0$ for $j = 1, 2, ..., m$, the inequality invariably holds. Then we set $\alpha_i^{(k)} = 1$. If the case that $g_i^{(k)} g_k^{(k)} < 0$, we can get:

$$\alpha_i^{(k)} \leq \frac{\alpha_k}{k-m} \frac{\|g_k^{(k)}\|^2}{g_i^{(k)} g_k^{(k)}}; \ k \leq i \leq m. \tag{18}$$

Since our target is to minimize the impact on other optimization objectives while avoiding anti-domination as much as possible, we equate the terms on both sides of Eq.18, ultimately deriving the solution:

$$\alpha_i^{(k)} = \begin{cases} 1 & \text{if} \quad g_i^{(k)} \cdot g_k^{(k)} \geq 0 \\ \frac{\alpha_k}{k-m} \frac{\|g_k^{(k)}\|^2}{g_i^{(k)} g_k^{(k)}} & \text{if} \quad g_i^{(k)} \cdot g_k^{(k)} < 0; k \leq i \leq m \end{cases} \tag{19}$$

Then, consider $\mathcal{L}_i$ and all $\mathcal{L}_k, k < i$, the $\alpha_i$ is as follows:

$$\alpha_i = min(\alpha_i^{(1)}, \alpha_i^{(2)}, ..., \alpha_i^{(i)}). \tag{20}$$

It is evident that the computational complexity of calculating $\alpha_i^{(k)}$ is $O(lb_k)$, where $l$ represents the dimension of the data feature $v$. Consequently, the overall computational complexity amounts to $O(lb_m m^2)$, where the $b_m$ is the longest code length.

## D   THE TRAINING ALGORITHM

In this section, we first present the training algorithm of our proposed NHL in Algorithm 1. Then, we elaborate on the details of our training process. As we discussed in Section 3.5, throughout the training procedure, we monitor the value of each $L_k$ and save the model parameters when each $L_k$ reaches its minimum to output the corresponding hash codes $h^{(k)}$. In lines 12-14 of Algorithm 1, when a smaller $L_k$ is achieved, We record the current model parameters $\theta_F$ and $\mathcal{W}$ as the parameters of the model with a length of $b_k$, denoted as $\theta_F^{(k)}$ and $\mathcal{W}^{(k)}$.

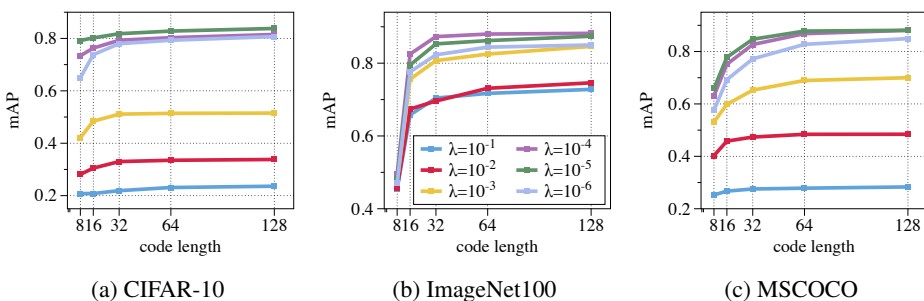

(a) CIFAR-10        (b) ImageNet100        (c) MSCOCO

Figure 7: The mAP@K results across three datasets under different learning rates.

Table 5: The $mAP@K$ evaluation when using different feature extraction networks as the backbone of CSQ across three datasets. We compared the results of using NHL (w/ NHL) with those of not using NHL (w/o NHL) on these three datasets.

| Data | Backbone | w/o NHL (The Original Model) | | | | | | w/ NHL | | | | | |
|---|---|---|---|---|---|---|---|---|---|---|---|---|---|
| | | 8 bits | 16 bits | 32 bits | 64 bits | 128 bits | avg. | 8 bits | 16 bits | 32 bits | 64 bits | 128 bits | avg. |
| CIFAR-10 (mAP@ALL) | ResNet18 | 0.654 | 0.710 | 0.743 | 0.717 | 0.739 | 0.712 | 0.664 | 0.718 | 0.762 | 0.793 | 0.817 | 0.750 (+5.36%) |
| | ResNet34 | 0.682 | 0.700 | 0.724 | 0.730 | 0.722 | 0.711 | 0.691 | 0.724 | 0.757 | 0.787 | 0.826 | 0.757 (+6.37%) |
| | ResNet50 | 0.762 | 0.786 | 0.798 | 0.798 | 0.807 | 0.790 | 0.792 | 0.802 | 0.818 | 0.828 | 0.838 | 0.816 (+3.21%) |
| | MobileNetV2 | 0.661 | 0.725 | 0.750 | 0.774 | 0.788 | 0.739 | 0.677 | 0.735 | 0.779 | 0.799 | 0.816 | 0.761 (+2.92%) |
| | ViT_B_16 | 0.891 | 0.903 | 0.907 | 0.912 | 0.913 | 0.905 | 0.895 | 0.909 | 0.919 | 0.931 | 0.937 | 0.918 (+1.44%) |
| ImageNet100 (mAP@1000) | ResNet18 | 0.375 | 0.727 | 0.754 | 0.773 | 0.795 | 0.684 | 0.413 | 0.722 | 0.774 | 0.804 | 0.815 | 0.705 (+3.03%) |
| | ResNet34 | 0.398 | 0.773 | 0.815 | 0.836 | 0.839 | 0.732 | 0.465 | 0.784 | 0.828 | 0.850 | 0.855 | 0.756 (+3.31%) |
| | ResNet50 | 0.456 | 0.822 | 0.860 | 0.877 | 0.878 | 0.778 | 0.495 | 0.825 | 0.873 | 0.880 | 0.882 | 0.787 (+1.59%) |
| | MobileNetV2 | 0.414 | 0.679 | 0.745 | 0.788 | 0.802 | 0.685 | 0.422 | 0.708 | 0.759 | 0.800 | 0.819 | 0.701 (+2.33%) |
| | ViT_B_16 | 0.523 | 0.888 | 0.910 | 0.915 | 0.915 | 0.830 | 0.533 | 0.887 | 0.906 | 0.918 | 0.925 | 0.834 (+0.43%) |
| MSCOCO (mAP@5000) | ResNet18 | 0.532 | 0.653 | 0.745 | 0.792 | 0.811 | 0.706 | 0.576 | 0.679 | 0.760 | 0.807 | 0.824 | 0.729 (+3.19%) |
| | ResNet34 | 0.554 | 0.715 | 0.789 | 0.830 | 0.836 | 0.744 | 0.591 | 0.731 | 0.802 | 0.840 | 0.847 | 0.762 (+2.33%) |
| | ResNet50 | 0.596 | 0.750 | 0.847 | 0.877 | 0.871 | 0.788 | 0.659 | 0.778 | 0.847 | 0.878 | 0.881 | 0.809 (+2.58%) |
| | MobileNetV2 | 0.543 | 0.668 | 0.742 | 0.806 | 0.823 | 0.716 | 0.578 | 0.691 | 0.766 | 0.815 | 0.836 | 0.737 (+2.90%) |
| | ViT_B_16 | 0.638 | 0.790 | 0.883 | 0.895 | 0.906 | 0.822 | 0.675 | 0.794 | 0.889 | 0.898 | 0.902 | 0.831 (+1.12%) |

## E MORE EXPERIMENTAL ANALYSES

### E.1 HYPER-PARAMETERS

We further conducted experimental validation on the CSQ model under different learning rate settings. We set the learning rates to $\{10^{-1}, 10^{-2}, 10^{-3}, 10^{-4}, 10^{-5}, 10^{-6}\}$ and performed experiments across three datasets. Figure 7 presents the results. Based on this experiment, we determined the optimal learning rate range to be $\{10^{-4}, 10^{-5}\}$.

### E.2 BACKBONE ANALYSIS

The NHL is designed for versatile integration with various deep supervised hashing models. To rigorously validate the generalizability of NHL, we extended this investigation by incorporating it into the CSQ deep hashing model while utilizing a diverse range of alternative backbone architectures for feature extraction. Specifically, we experimented with ResNet variants of different scales, including ResNet18 and ResNet34. We also present the results of ResNet50 used previously for comparison. Furthermore, to assess NHL's compatibility with distinct network structures, we also employed the Vision Transformer (ViT_B_16) (Dosovitskiy et al., 2020) and the lightweight MobileNetV2 architecture (Sandler et al., 2018). As demonstrated in Table 5, NHL consistently maintains its effectiveness when paired with these varied backbones. This underscores NHL's broad applicability and its robustness in enhancing hashing performance across different types and scales of feature extractors, from various CNNs to transformer-based models.

## F ABLATION STUDY

This section presents the comprehensive results of NHL variants applied to deep hashing models. Figures 8, 6, and 7 showcase the outcomes for CIFAR-10, ImageNet100, and MSCOCO, respectively. The best results are highlighted in bold, while the second-best results are underlined. From these

results, it is evident that in the vast majority of cases, employing the complete NHL method achieves optimal performance. Additionally, in certain scenarios, using variants of NHL yields the best results, which can be attributed to the inherent differences among various deep hashing models, as well as the influence of different hash code lengths and datasets. As a plug-and-play module, NHL demonstrates sufficient robustness and adaptability across diverse applications.

Table 6: The mAP@K comparison results on the ImageNet100 dataset when different deep hashing models use the original hash layer, NHL, or the variants of NHL. The best results are highlighted in bold, while the second-best results are underlined.

| Data | Length | NHL | DSH | DTSH | LCDSH | DCH | CSQ | DPN | SHCIR | MDSH |
|---|---|---|---|---|---|---|---|---|---|---|
| ImageNet100 (mAP@1000) | 8bit | w/o | 0.703 | 0.432 | 0.248 | 0.776 | 0.456 | 0.436 | 0.789 | 0.785 |
| | | -basic | 0.630 | 0.544 | 0.407 | 0.809 | 0.424 | 0.453 | 0.744 | 0.697 |
| | | w/o D | 0.688 | 0.512 | 0.413 | **0.810** | 0.451 | 0.456 | 0.764 | 0.787 |
| | | w/o L | 0.663 | 0.535 | **0.426** | 0.797 | 0.487 | 0.453 | 0.777 | 0.756 |
| | | w/ | **0.755** | **0.552** | 0.422 | 0.809 | **0.495** | **0.487** | 0.798 | **0.794** |
| | 16bit | w/o | 0.808 | 0.710 | 0.395 | 0.834 | 0.822 | 0.827 | 0.861 | 0.845 |
| | | -basic | 0.787 | 0.682 | 0.571 | 0.845 | 0.781 | 0.830 | 0.831 | 0.812 |
| | | w/o D | 0.797 | 0.701 | **0.597** | **0.846** | 0.803 | **0.832** | 0.822 | 0.836 |
| | | w/o L | 0.784 | **0.721** | 0.554 | 0.839 | 0.820 | 0.825 | 0.828 | 0.827 |
| | | w/ | **0.816** | 0.714 | 0.568 | 0.842 | **0.825** | 0.829 | **0.881** | **0.851** |
| | 32bit | w/o | 0.827 | **0.770** | 0.542 | 0.845 | 0.860 | **0.864** | 0.879 | 0.874 |
| | | -basic | 0.814 | 0.765 | 0.622 | 0.856 | 0.847 | 0.862 | 0.867 | 0.859 |
| | | w/o D | 0.808 | 0.721 | **0.645** | **0.858** | 0.846 | 0.861 | 0.867 | 0.846 |
| | | w/o L | 0.807 | 0.749 | 0.601 | 0.850 | 0.848 | 0.858 | 0.861 | 0.860 |
| | | w/ | **0.829** | 0.766 | 0.628 | 0.855 | **0.873** | 0.860 | **0.889** | **0.878** |
| | 64bit | w/o | 0.828 | 0.784 | 0.608 | 0.859 | 0.877 | 0.870 | 0.883 | **0.895** |
| | | -basic | 0.825 | 0.773 | 0.671 | **0.864** | 0.874 | 0.875 | 0.881 | 0.879 |
| | | w/o D | 0.818 | 0.760 | **0.679** | 0.863 | 0.873 | 0.874 | 0.885 | 0.872 |
| | | w/o L | 0.820 | 0.769 | 0.631 | 0.854 | 0.864 | 0.870 | 0.882 | 0.879 |
| | | w/ | **0.838** | **0.788** | 0.657 | 0.860 | **0.880** | **0.877** | **0.893** | 0.884 |
| | 128bit | w/o | 0.822 | **0.803** | 0.692 | 0.848 | 0.878 | 0.877 | 0.881 | 0.894 |
| | | -basic | 0.827 | 0.782 | 0.703 | 0.863 | **0.888** | 0.878 | 0.887 | 0.888 |
| | | w/o D | 0.820 | 0.764 | **0.714** | **0.866** | 0.884 | 0.880 | 0.884 | 0.883 |
| | | w/o L | 0.825 | 0.770 | 0.665 | 0.857 | 0.878 | 0.877 | 0.887 | 0.886 |
| | | w/ | **0.841** | 0.792 | 0.692 | 0.863 | 0.882 | **0.881** | **0.898** | **0.896** |

Table 7: The $mAP@K$ comparison results on the MSCOCO dataset when different deep hashing models use the original hash layer, NHL, or the variants of NHL. The best results are highlighted in bold, while the second-best results are underlined.

| Data | Length | NHL | DSH | DHN | DTSH | LCDSH | DCH | DBDH | CSQ | DPN |
|---|---|---|---|---|---|---|---|---|---|---|
| MSCOCO (mAP@1000) | 8bit | w/o | 0.685 | 0.659 | 0.706 | 0.687 | 0.695 | 0.655 | 0.596 | 0.575 |
| | | -basic | 0.693 | 0.714 | 0.737 | 0.708 | 0.720 | 0.676 | 0.635 | 0.621 |
| | | w/o D | 0.662 | 0.705 | 0.738 | **0.727** | 0.721 | 0.690 | 0.638 | 0.633 |
| | | w/o L | 0.688 | 0.708 | 0.734 | 0.703 | 0.692 | 0.688 | 0.636 | **0.646** |
| | | w/ | **0.714** | **0.724** | **0.751** | 0.713 | **0.723** | 0.692 | **0.659** | 0.638 |
| | 16bit | w/o | 0.722 | 0.751 | 0.770 | 0.769 | 0.756 | 0.727 | 0.750 | 0.757 |
| | | -basic | 0.730 | 0.757 | 0.769 | 0.757 | 0.768 | 0.732 | 0.735 | 0.747 |
| | | w/o D | 0.705 | 0.751 | 0.775 | 0.771 | **0.773** | 0.737 | 0.769 | **0.779** |
| | | w/o L | 0.726 | **0.764** | 0.789 | 0.761 | 0.736 | 0.747 | 0.749 | 0.774 |
| | | w/ | **0.735** | 0.760 | **0.793** | **0.773** | 0.769 | 0.748 | 0.778 | 0.769 |
| | 32bit | w/o | 0.757 | 0.786 | 0.810 | 0.787 | 0.762 | 0.760 | **0.847** | 0.828 |
| | | -basic | 0.747 | 0.789 | 0.791 | 0.774 | **0.788** | 0.764 | 0.822 | 0.820 |
| | | w/o D | 0.731 | 0.783 | 0.805 | 0.788 | 0.787 | 0.765 | 0.822 | 0.835 |
| | | w/o L | 0.749 | 0.792 | 0.809 | 0.789 | 0.757 | **0.781** | 0.811 | 0.819 |
| | | w/ | **0.764** | **0.794** | **0.819** | **0.794** | 0.786 | 0.778 | **0.847** | **0.837** |
| | 64bit | w/o | 0.779 | 0.810 | **0.823** | 0.825 | 0.777 | 0.769 | 0.877 | 0.862 |
| | | -basic | 0.765 | 0.809 | 0.796 | 0.780 | **0.791** | 0.783 | 0.866 | 0.853 |
| | | w/o D | 0.749 | 0.804 | 0.808 | 0.798 | 0.789 | 0.783 | 0.866 | 0.860 |
| | | w/o L | 0.765 | 0.815 | 0.814 | 0.826 | 0.765 | 0.803 | 0.857 | 0.849 |
| | | w/ | **0.789** | **0.837** | 0.823 | **0.828** | 0.789 | **0.809** | **0.881** | **0.872** |
| | 128bit | w/o | 0.769 | 0.832 | **0.831** | 0.836 | 0.734 | 0.800 | 0.871 | 0.863 |
| | | -basic | 0.773 | 0.820 | 0.797 | 0.786 | 0.787 | 0.799 | 0.876 | 0.868 |
| | | w/o D | 0.761 | 0.822 | 0.814 | 0.802 | 0.787 | 0.797 | 0.877 | **0.872** |
| | | w/o L | 0.774 | 0.828 | 0.815 | **0.839** | 0.764 | **0.811** | 0.873 | 0.866 |
| | | w/ | **0.789** | **0.837** | 0.823 | 0.828 | **0.789** | 0.809 | **0.881** | **0.872** |

Table 8: The mAP@K comparison results on the CIFAR-10 dataset when different deep hashing models use the original hash layer, NHL, or the variants of NHL. The best results are highlighted in bold, while the second-best results are underlined.

| Data | Length | NHL | DSH | DHN | DTSH | LCDSH | DCH | DBDH | CSQ | DPN | SHCIR | MDSH |
|---|---|---|---|---|---|---|---|---|---|---|---|---|
| CIFAR-10 (mAP@1000) | 8bit | w/o | 0.690 | 0.718 | 0.754 | 0.715 | 0.776 | 0.737 | 0.762 | 0.703 | 0.754 | 0.755 |
| | | -basic | 0.598 | 0.724 | 0.560 | 0.759 | 0.779 | 0.731 | 0.652 | **0.731** | **0.761** | 0.752 |
| | | w/o D | 0.620 | 0.730 | **0.772** | 0.751 | 0.772 | **0.750** | 0.707 | 0.727 | 0.740 | 0.743 |
| | | w/o L | 0.641 | **0.752** | 0.747 | 0.773 | 0.781 | 0.737 | 0.769 | 0.632 | 0.760 | 0.760 |
| | | w/ | **0.717** | 0.739 | 0.766 | **0.775** | **0.787** | 0.748 | **0.792** | 0.729 | 0.758 | **0.762** |
| | 16bit | w/o | 0.731 | 0.765 | 0.778 | 0.771 | 0.802 | 0.771 | 0.786 | 0.757 | 0.791 | 0.808 |
| | | -basic | 0.688 | 0.778 | 0.702 | 0.782 | 0.806 | 0.769 | 0.758 | 0.768 | 0.796 | 0.788 |
| | | w/o D | 0.667 | 0.769 | 0.785 | 0.766 | 0.798 | 0.770 | 0.773 | **0.783** | 0.777 | 0.775 |
| | | w/o L | 0.710 | **0.797** | 0.775 | 0.785 | 0.808 | 0.764 | 0.777 | 0.752 | 0.793 | 0.801 |
| | | w/ | **0.732** | 0.776 | **0.790** | **0.799** | **0.810** | **0.785** | **0.802** | 0.765 | **0.797** | **0.811** |
| | 32bit | w/o | 0.740 | 0.813 | 0.799 | 0.817 | 0.829 | 0.796 | 0.798 | 0.790 | 0.820 | 0.829 |
| | | -basic | 0.726 | 0.818 | 0.777 | 0.815 | 0.827 | 0.797 | 0.795 | 0.804 | **0.834** | 0.823 |
| | | w/o D | 0.733 | 0.783 | **0.814** | 0.788 | 0.822 | 0.799 | 0.815 | **0.812** | 0.812 | 0.814 |
| | | w/o L | 0.739 | 0.815 | 0.796 | 0.799 | 0.828 | 0.793 | 0.803 | 0.792 | 0.828 | 0.831 |
| | | w/ | **0.744** | **0.824** | 0.802 | **0.825** | **0.833** | **0.804** | **0.818** | 0.795 | 0.824 | **0.838** |
| | 64bit | w/o | 0.727 | **0.837** | **0.831** | 0.826 | 0.830 | **0.829** | 0.798 | 0.804 | 0.844 | 0.844 |
| | | -basic | 0.734 | 0.834 | 0.810 | 0.823 | 0.834 | 0.822 | 0.812 | 0.819 | 0.853 | 0.845 |
| | | w/o D | 0.740 | 0.802 | 0.820 | 0.803 | 0.827 | 0.811 | 0.823 | 0.817 | 0.834 | 0.838 |
| | | w/o L | 0.739 | 0.833 | 0.808 | 0.814 | 0.835 | 0.816 | 0.821 | 0.819 | 0.848 | 0.849 |
| | | w/ | **0.743** | 0.835 | 0.822 | **0.839** | **0.843** | 0.825 | **0.828** | **0.826** | **0.849** | **0.852** |
| | 128bit | w/o | 0.381 | 0.853 | 0.811 | 0.854 | 0.825 | 0.829 | 0.807 | 0.819 | 0.828 | 0.832 |
| | | -basic | **0.752** | 0.846 | 0.835 | 0.835 | 0.829 | 0.838 | 0.824 | 0.824 | 0.827 | 0.852 |
| | | w/o D | 0.744 | 0.821 | 0.831 | 0.824 | 0.831 | 0.834 | 0.831 | 0.821 | 0.838 | 0.845 |
| | | w/o L | 0.736 | 0.852 | 0.818 | 0.828 | 0.839 | **0.873** | 0.832 | **0.827** | **0.857** | 0.858 |
| | | w/ | 0.749 | **0.866** | **0.836** | **0.868** | **0.844** | 0.844 | **0.838** | 0.824 | 0.848 | **0.861** |

## G  LLM USAGE STATEMENT

In compliance with the ICLR 2026 policy, we disclose the use of a large language model as an assistive tool in the preparation of this manuscript.

The model used was **Gemini 2.5-Pro**. Its role was strictly limited to that of a writing assistant for polishing parts of the text. Specifically, it was used to improve grammar, clarity, and conciseness for author-written content. The LLM was not used for core research ideation, experimental design, data analysis, or the formulation of our conclusions.

Our workflow for using the LLM followed a strict three-step, human-in-the-loop process:

1. **Polish:** We used the model to suggest alternative phrasing or grammatical corrections for existing text drafted by the authors.

2. **Review:** All suggestions provided by the LLM were critically reviewed by the authors to verify their accuracy and to ensure they did not alter the original scientific meaning or intent.

3. **Manual Revision:** We manually integrated and modified any useful suggestions to ensure the final text accurately and precisely reflected our findings and narrative.

The authors take full responsibility for all content presented in this paper, including any text that was revised with the assistance of the LLM.

