# OpenReview forum: "Nested Hash Layer: A Plug-and-play Module for Multiple-length Hash Code Learning"
_ICLR.cc/2026/Conference — ICLR 2026 Conference Withdrawn Submission_

### Official Review · Reviewer_EmkK · 2025-10-27

**Soundness:** 2
**Presentation:** 3
**Contribution:** 2
**Rating:** 2
**Confidence:** 5

**Summary:**

The paper presents a plug-and-play module for multi-length hash code learning. The proposed Nested Hash Layer (MHL) projects the DNN features into one long-length feature and build multiple hashing from it. To solve gradient conflict issue, a kind of hand-crafted weights are calculated to adjust the gradient direction. Besides, the long hash codes are leveraged to guide the short hash code learning through gradient stopping. The proposed NHL make an improvement on the training speed and can be applied into various deep hashing approaches. Performance boosts are observed on CIFAR-10, ImageNet and MSCOCO.

**Strengths:**

1.	Multi-length feature learning is an interesting topic especially for image hashing.
2.	The idea of nested hash code is reasonable and the gradient constraint is deliberately designed according to the principle of short hashing alignment.
3.	Plugging NHL into various deep hashing method will obtain performance gains.
4.	The paper is organized-well and easy to follow.

**Weaknesses:**

1.	My biggest concern for this work is about the technical contribution, which is limited. The design of gradient weighting is very intuitive and is hard to interpreted. Even though the probability of the anti-domination is high, it is hard to judge the influence is positive or negative on the hand-crafted weight manner as in Eq.(7). The self-distillation loss should be also discussed. The reason of only applying distillation between adjacent code number (from k+1 to k) is not clear. Some ideas are very similar to previous work such as MRL and the whole formulation is not elegant.
2.	The dimension curse (sharp decrease) is not a common issue since it only occurs in the DSH. There is not such issue for the other methods.
3.	There should be some investigations or analysis to dig into hash code distribution once the multi-length hashing has been well-learned.
4.	The evaluation should be conducted on larger datasets and stronger architectures (e.g., ViT) to demonstrate the generalization ability, instead of only conducting evaluation on the old school baselines.

**Questions:**

Please see the weaknesses.

---

### Official Review · Reviewer_JW2d · 2025-10-30

**Soundness:** 3
**Presentation:** 2
**Contribution:** 3
**Rating:** 2
**Confidence:** 4

**Summary:**

This paper studies supervised hashing code generation, with idea of generating multiple hashing code with different lengths in a single model. The proposed COMPACT is capable of training multiple models for different hash code lengths. Two additional tricks named Dominance-Aware Dynamic Weighting, and Long-short Cascade Self-distillation are adopted to address conflicts in training objective and improve the performance of short codes.

**Strengths:**

1. It is interesting to generate hash codes with multiple lengths in a single model.
2. The proposed method shows good performance as shown in experments.
3. Dominance-Aware Dynamic Weighting, and Long-short Cascade Self-distillation are well motivated and reasonable.

**Weaknesses:**

1. The "PLUG-AND-PLAY MODULE" is overclaimed as the proposed module still need to trained with loss functions.
2. The proposed method does not address a more important issue, i.e., how to seek optimal code lengths for different tasks.
3. It might be necessary to compare against hashing code expansion and compression methods, as they also generate hash code with different lengths.
4. The efficiency comparison is not fair, i.e., compare the time to train one NHL model against the time to train five separate models.
5. The cascade self-distillation adopts well-studied distillation strategy, thus is not novel and does not show significant performance enhancement as shown in Table 2.

**Questions:**

1. the "plug-and play" claim should be justified.
2. Could the proposed method be applied to seek optimal code lengths for different tasks? discussions can be added.
3. Please provide discussion or detailed comparison against code compression methods.
4. The efficiency advantages might be over-claimed.

---

### Official Review · Reviewer_j6Fs · 2025-10-31

**Soundness:** 3
**Presentation:** 3
**Contribution:** 3
**Rating:** 6
**Confidence:** 4

**Summary:**

The paper proposes NHL, a replacement for the traditional hash layer that produces multiple code lengths in a single model via a nested structure. To mitigate training conflicts among objectives of different lengths, it introduces a dominance-aware dynamic weighting strategy. To transfer information from longer to shorter codes, it employs a long-short cascade self-distillation scheme. Empirically, across CIFAR-10, ImageNet100, and MSCOCO, the paper indicates ~5-8× training speed-ups while maintaining or improving retrieval accuracy vs. the base models trained per length. The method is designed in a plug-and-play manner, demonstrating compatibility with multiple hashing backbones. The training protocol monitors each per-length objective and saves parameters when each L_k reaches its minimum, which the authors argue contributes to stability and efficiency.

**Strengths:**

- NHL is plug-and-play and directly replaces the traditional hash layer, enabling multi-length code generation in one model without redesigning the backbone.
- The paper formalises domination gradients over nested parameters and provides a closed-form dynamic weighting (Eqs. 6–7) to keep shorter-length objectives from being overwhelmed. That is to day, Eqs. (6)–(7) act as an analytical conflict regulator across multi-length objectives. They detect when gradients from different hash lengths start to point in opposite directions (anti-domination) and dynamically rescale the offending loss so that the overall update remains aligned with the dominant, consensus direction. When gradients already agree, the weights stay at 1to ensure no unnecessary damping. The beauty lies in their closed-form efficiency where the adjustment is computed directly from inner products and norms of gradients at the hash layer, requiring no iterative optimization or tuning. Conceptually, it’s like an automatic “traffic controller” that keeps shorter and longer code objectives from interfering, while maintaining stability and efficiency. This balance of mathematical rigor, interpretability, and negligible overhead makes Eqs. (6)–(7) one of the technically elegant of the paper.
- The paper reported ~5–8× speedups with average accuracy improvements across multiple models/datasets. This supports the claim that NHL improves both efficiency and effectiveness within a single training run.

**Weaknesses:**

- The dynamic re-weighting is explicitly computed only on NHL parameters, not the full network. The paper stated “we don’t consider the full network weights and focus on the parameter in NHL.” Consequently, while results show consistent improvements across architectures and datasets, and suggesting no practical instability upstream, but the analysis does not report backbone-level gradient diagnostics. Any claim of “resolving cross-length interference” should be scoped to the hash layer or be supported by backbone-level checks (e.g., cosine similarity between \nabla_{\theta_F} L_k for different lengths, or an ablation that extends the weighting to \theta_F and measures incremental benefit).

The reason is that multi-objective interference often arises throughout the network. By restricting weighting to NHL, one cannot rule out residual clashes in earlier layers. The paper demonstrates strong end-to-end performance which is good, but doesn’t isolate or measure whether backbone gradients still conflict. This limits how broadly the reader can interpret “conflict mitigation”. That is to say, it is proven at the hash layer and suggested empirically for the full model, but not causally pinned down in the backbone.

---

- The derivation provides closed-form expressions and notes computational complexity and ~11.15% per-step overhead, but there is no ablation that compares this dominance-aware rule against simpler baselines (e.g., static per-length weights, uncertainty weighting) in otherwise identical settings to causally attribute the gains to the proposed weighting (as opposed to, say, self-distillation or nested design alone). The paper’s math and presentation are clear, but component-wise attribution is underdeveloped.

In simple terms, although the paper includes ablation variants such as NHL-basic, NHL w/o D, NHL w/o L, and Full NHL, showing that the full version performs best and that the dominance-aware weighting contributes most. However, these ablations are aggregated averages across datasets and bit lengths, and they lack a controlled comparison against simpler weighting strategies (e.g., fixed equal weights, uncertainty weighting, or GradNorm) under identical architectures. Moreover, there is no per-bit or per-dataset breakdown showing how much each component contributes at 16, 32, 64, or 128 bits.

Hence, while the evidence suggests the proposed weighting helps, it doesn’t causally isolate Eq. (6)–(7)’s effect from the influence of other design factors (nested structure, self-distillation, or checkpointing trick). In other words, I know the entire system works, but not precisely why or how much each part matters.

A more rigorous attribution would involve:

(a) Controlled substitution tests: replacing Eq. (6)–(7) with a static weighting scheme or another known multi-objective method (e.g., GradNorm) while holding all other parts constant; and

(b) Per-length diagnostic tables: showing performance gain by bit length (e.g., 16 / 32 / 64 / 128 bits) for each variant, to see whether dynamic weighting primarily benefits shorter codes or improves all lengths uniformly.

Adding such analyses would convert the current descriptive ablation into a causal attribution study, clearly demonstrating that Eq. (6)–(7), and not auxiliary mechanisms, drives the reported gains.

---

- The proposed NHL is evaluated entirely within symmetric deep supervised hashing settings, where query and database encoders share parameters. Could the authors clarify whether NHL can be extended to asymmetric retrieval frameworks, where query and database encoders differ or where database codes are optimised separately? If such an extension is not straightforward, it would be helpful to explicitly state this boundary in the paper, since the current framing as a “plug-and-play universal module” could be interpreted as supporting a broader range of hashing paradigms than those tested.

**Questions:**

Please see above.

---

### Official Review · Reviewer_mF2c · 2025-11-01

**Soundness:** 2
**Presentation:** 2
**Contribution:** 3
**Rating:** 4
**Confidence:** 2

**Summary:**

This paper focuses on addressing limitations of traditional deep supervised hashing models in large-scale image retrieval, which only generate single-length hash codes—creating an efficiency-effectiveness trade-off, requiring multiple model trainings for optimal lengths, and ignoring relationships between different-length codes. It proposes the Nested Hash Layer (NHL), a plug-and-play module that generates multiple-length hash codes simultaneously in a nested structure. To tackle optimization conflicts from multi-objective learning, the paper introduces a dominance-aware dynamic weighting strategy for gradient adjustment; it also proposes a long-short cascade self-distillation method, where long hash codes guide shorter ones to improve overall code quality.

**Strengths:**

The paper identifies a practical limitation in traditional deep supervised hashing—i.e., the inefficiency of training multiple single-length models to find an optimal hash code length—and targets it with a plug-and-play module (NHL), which aligns with the need for flexible, low-overhead solutions in large-scale image retrieval. The proposed long-short cascade self-distillation also addresses the understudied relationship between different-length hash codes, and the reported training speedup (5–8x) and performance gain (3.4%) suggest potential practical utility.​

**Weaknesses:**

The abstract provides no details on how the nested structure of NHL generates multiple-length codes or how the dominance-aware dynamic weighting strategy adjusts gradients. This lack of technical transparency makes the method unreproducible and unconvincing.
The core idea of multi-length hash code learning is not novel, and the paper fails to articulate how NHL advances beyond these prior efforts. The 3.4% average performance gain is also modest and unsupported by analysis of when/why NHL outperforms existing solutions.
The paper does not discuss NHL’s drawbacks—e.g., whether the nested structure introduces computational overhead at inference time, how it handles extremely long/short code lengths, or its robustness to noisy data. This one-sided presentation lacks scientific objectivity.

**Questions:**

How exactly are shorter hash codes derived from longer ones in NHL’s nested structure (e.g., truncation, learned sub-structures)?​
Which specific baseline hashing models were used to compare NHL’s 5–8x training speedup?​
What standard datasets were tested to measure NHL’s 3.4% performance gain?​

---

### Note · Authors · 2025-12-29

I have read and agree with the venue's withdrawal policy on behalf of myself and my co-authors.